# Nozzle Geometry and Particle Size Influence on the Behavior of Low Pressure Cold Sprayed Hydroxyapatite Particles

Paola Andrea Forero-Sossa [1], Astrid Lorena Giraldo-Betancur [2], Carlos A. Poblano-Salas [1], Aixa Ibeth Gutierrez-Pérez [3], Esaú Moises Rodríguez-Vigueras [1], Jorge Corona-Castuera [1] and John Henao [4,*]

1 CIATEQ A.C., Av. Manantiales 23-A, Parque Industrial Bernardo Quintana, El Marqués, Queretaro 76246, Mexico
2 CONACyT-Centro de Investigación y de Estudios Avanzados del IPN, Unidad Querétaro, Libramiento Norponiente, #2000, Querétaro 76230, Mexico
3 Centro de Investigación y de Estudios Avanzados del IPN, Unidad Querétaro, Libramiento Norponiente, #2000, Queretaro 76230, Mexico
4 CONACyT-CIATEQ A.C., Av. Manantiales 23-A, Parque Industrial Bernardo Quintana, El Marqués, Queretaro 76246, Mexico
* Correspondence: john.henao@ciateq.mx

**Abstract:** Low-pressure cold spray (LPCS) technology has attracted interest for the deposition of ceramic coatings due to the thermo-kinetic conditions experienced by the sprayed particles. Unlike conventional thermal spray techniques, the spraying conditions in LPCS can be controlled to avoid the formation of undesired phases. However, ceramics deposition through this process is still challenging. The present study includes a finite element analysis and simulation study of the kinetic conditions of ceramic particles in the LPCS process based on experimental data. The analysis seeks to discuss the effect of nozzle geometry on the kinetic and thermal energy of the sprayed particles at impact and elucidate how the particle travels within the high-velocity jet to be deposited onto a metallic surface. This work examines the behavior of hydroxyapatite particles as a function of particle size and nozzle geometry during LPCS deposition. Interestingly, the results from this research suggest that particle size and nozzle geometry have an influence on the deposition of hydroxyapatite particles. Inertia of large particles proved to be beneficial in keeping their trajectories, allowing them to contribute to the formation of the coatings. Nozzle geometry modifications produced changes in the jet profile and affected the homogeneity of the coatings obtained. This finding contributes to a better understanding of the deposition of hydroxyapatite particles by cold spraying.

**Keywords:** cold spray; ceramics; bovine-derived hydroxyapatite; low pressure; nozzle geometry

## 1. Introduction

Cold spraying (CS) is a thermal spray technique developed in the 1980s by Dr. Anatolli Papyrin and his collaborators at the Institute of Theoretical and Applied Mechanics of the Russian Academy of Sciences (ITAM-RAS) [1]. CS is a kinetic-based process, as the particles can reach speeds between 300 and 1600 m/s and temperatures below 1100 °C; therefore, the particle deposition behavior in CS relates mainly to the nature of the material and its mechanical response due to the impact at high velocity [2,3]. Depending on the operating pressure range, CS systems are classified into two main categories: high-pressure cold spray (HPCS) and low-pressure cold spray (LPCS). HPCS operates at maximum gas pressures up to 60 bar (6.0 MPa), whereas the working gas temperature lies in the range of 300 to 1100 °C and particle velocities are often in the supersonic regime. On the other hand, the LPCS system operates at maximum pressures of 10 bar (1 MPa) with gas temperatures in the range of 20 to 650 °C and particle velocities typically below 600 m/s [4].

CS has been successfully used to deposit metallic materials, such as copper, nickel, and aluminum, among others [2]. The coatings obtained by this technique can have thicknesses

of up to a few millimeters with porosities less than 1% and deposition efficiencies close to 100% [5]. Different theoretical models can predict the processing conditions that favor the formation of a coating in CS, known as the deposition window. One of the models that predict the behavior of metal particles by CS is the so-called critical velocity model proposed by Schmidt et al. [2]; this model considers the characteristics and properties of the materials to be deposited. Critical velocity is known in metals to describe the starting limit from which deposition efficiency increases and coating growth is expected. On the other hand, computational fluid dynamics (CFD) and numerical models have been previously employed along with experimental observations to understand the phenomena that affect metal particles' impact on metallic substrates. One of the most widely accepted models in the scientific community regarding the growth and formation of metallic coatings by CS is the so-called adiabatic shear instability model proposed by Assadi et al. [6]. The model involves plastic deformation of metals. Other materials, such as metallic glasses, have also been studied by CS using CFD. For metallic glasses, the success of deposition and coating formation estimations have been associated with the Reynolds number of the particles at impact [7], which involves their viscous-plastic deformation.

In contrast, CS deposition of ceramic materials is still a state-of-the-art process. The fragile behavior and lack of ductility of ceramic materials in the solid state are well-known. Despite this, some authors have reported the feasibility of preparing ceramic coatings by CS, such as titanium oxide ($TiO_2$) and hydroxyapatite (HAp), with thicknesses of up to 300 μm and 120 μm, respectively. Interestingly, $TiO_2$ coatings have been obtained by CS processes. Overall, the success in obtaining $TiO_2$ coatings by CS is associated with the morphology and size distribution of the particles and the atomic arrangement of oxygen and titanium. For instance, the thickest $TiO_2$ coatings obtained by HPCS are associated with agglomerated and submicrometric particles having a tetragonal crystal structure [8]. On the other hand, the thickest coatings produced by LPCS are associated with agglomerated and micrometric particles (ranging from 10 and 50 μm) having an amorphous structure [9]. Concerning HAp coatings by CS, Chen et al. obtained thicknesses of 120 μm using an HPCS system and a special nozzle designed and manufactured by his group [10]. Moreover, Cinca et al. [11] reported that LPCS systems may allow the fabrication of thick HAp coatings. Most of the previous studies working with $TiO_2$ and HAp associate the deposition of these materials with particle size and particle morphology. However, the processing conditions and critical velocity required for ceramic materials deposition by CS have been little discussed.

Nowadays, the deposition mechanism of ceramic materials by CS is still at the frontier of knowledge. No model can fully explain and predict a deposition window that guarantees the success of building up ceramic coatings, as currently happens with metallic materials. Yet, some approximations and proposals are looking to explain such mechanism. For instance, Yamada et al. [12] suggests that the formation of the coating is due to the occurrence of a chemical bond between the particles and the substrate; whereas Cinca et al. [13] proposed a so-called pore collapse model for agglomerated particles, which consists of three stages: (i) slight collapse of the pores present in the HAp particles, (ii) dynamic fragmentation produced by cracking and crushing, and (iii) reduction of the crystallite size [10,13]. Recently, Chakrabarty et al. [14] suggested that fracture of the particles can reduce deposition efficiency due to the rebounding experienced by the fragments upon impact. In this sense, Yashima et al. [15] reported that ceramic particles' fracture speed increases as the particle size decreases.

In recent years, the thermal spray industry has acquired a growing interest in ceramics behavior when deposited using the CS technique. The reason is that cold spray is a very attractive technique when physical and chemical characteristics of the feedstock powder are to be maintained in the final coatings. Ceramic coatings obtained by cold spray might have multiple applications as sensors, biomedical devices, thermal barriers, and so on. However, to date, ceramic coatings produced by cold spray seem to be more of a desire than a fact due to the lack of in-depth studies of deposition conditions of these types of materials by this technique.

The present study seeks to contribute to the understanding of the kinetic conditions achieved by HAp particles when sprayed by LPCS. In addition, this study aims to obtain new experimental observations that may help to describe the deposition of HAp coatings under solid-state impact conditions. The results can be contrasted with those of previous experimental works and may be of interest for future experimental applications in the field of biomedical coatings.

## 2. Materials and Methods

A LPCS equipment (DYMET 423) was used to study the gas dynamics and kinetic behavior of hydroxyapatite (HAp) particles (See Supplementary Material, Figure S1). One of the most important parameters of the LPCS process is the nozzle geometry, since it has an influence on the compression and expansion of the gas. In this work, two nozzle geometries were employed to evaluate the behavior of the gas and particles: (i) a commercially available round nozzle (CK20, DYCOMET, Akkrum, Netherlands) named as "conventional", with a divergent length of 120 mm and an expansion ratio of 5.32 (Figure 1a) [16]. This conventional nozzle has a convergent-divergent type geometry characterized by the following sections: (1) inlet, (2) prechamber, (3) convergent, (4) throat, (5) divergent, and (6) outlet; and (ii) a homemade nozzle labeled as "modified", which was manufactured and evaluated to compare gas and particle behavior with the conventional counterpart. This new nozzle has a total length of 17.7 mm longer than the conventional one. The dimensional changes in the modified nozzle were focused on the throat length (4) and convergent section (3). In particular, a trumpet-type convergent section (3) was considered using a compressive ratio of 1.32, as suggested in previous experimental reports [10]. The modified nozzle geometry is shown in Figure 1a. This nozzle was fabricated by additive-manufacturing employing the direct melting laser sintering process (DMLS, EOS M280) and Inconel 625 spherical powder. The printing parameters used for the preparation of the modified nozzle are those suggested by the powder's manufacturer and are reported elsewhere [17,18]. Once the nozzle was 3D-printed, conventional machining was performed to obtain the threads that are required for attaching the nozzle to the cold spray gun (DYCOMET, Akkrum, The Netherlands).

Computational fluid dynamics (CFD) was employed to model the gas dynamics and particle thermo-kinetics behavior inside the convergent-divergent nozzles used in this work. ANSYS-FLUENT® software (version 2021 R1 Academic) was employed to perform modelling in all cases. Air was used as propelling gas and modeled under the ideal gas concept, which considers compressibility effects and changes on gas density. The supersonic flow model available in the employed software was based on the simultaneous solution of the momentum, mass, and energy conservation equations, similar to previous studies [19,20]. Also, the density-based solver method was used for gas simulations, as the compressibility phenomenon approximates better than the pressure-based resolution method in LPCS systems [21].

The gas-particle interaction was modeled according to the Basset-Boussinesq-Ossen simplified equation and following the discrete phase model (DPM), which allows studying granular flows. The physical and mechanical properties of HAp considered in this study are presented in Table 1. The injection of a single HAp particle of different diameters was evaluated in the simulations. These diameters were established based on the experimental particle size distribution obtained for bovine-derived hydroxyapatite (BHAp) powder (see Supplementary Material, Figure S2). Furthermore, HAp particles were considered inert and experienced only heat transfer and turbulence phenomena, which is a typical behavior in the CS technique [20,22]. Particles were injected radially at the end of the throat, as is usual in the LPCS system. Figure 1b presents the injection point of the powder in the nozzle (see red arrow). The properties used for the gas are also summarized in Table 1.

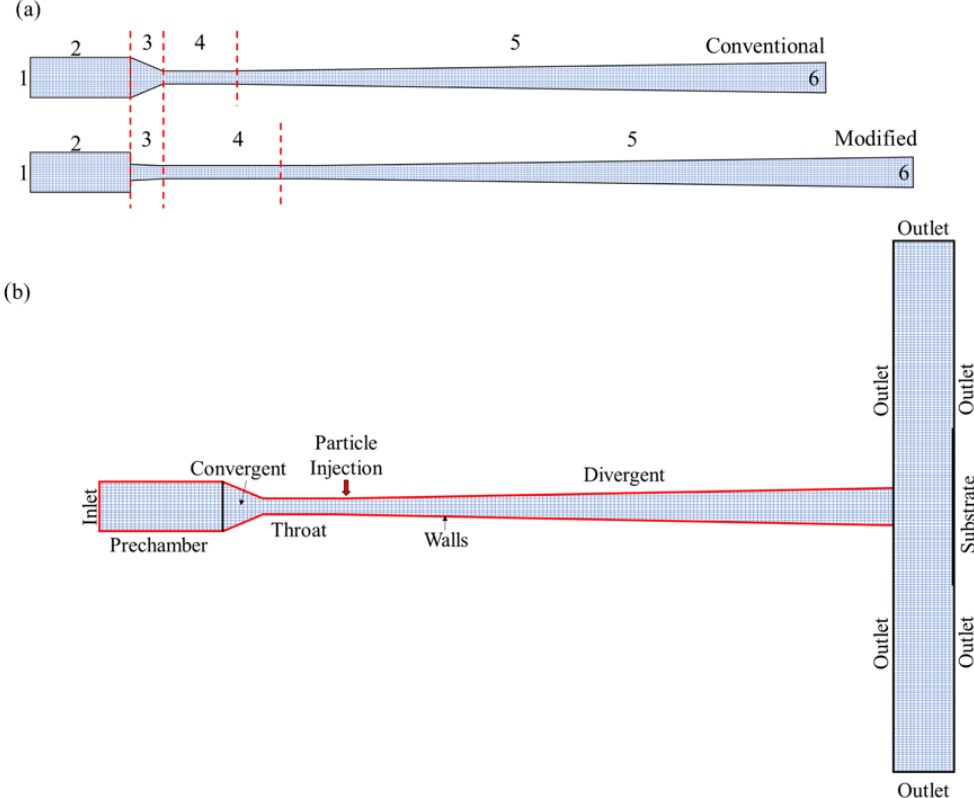

**Figure 1.** (**a**) Scale drawing of conventional and modified nozzle geometry. Nozzle sections are numbered as follows: (1) inlet, (2) prechamber, (3) converging, (4) throat, (5) diverging, and (6) outlet; (**b**) Sections and boundary conditions used in the conventional nozzle.

**Table 1.** Gas and particles properties used for CFD simulations.

| Property | Gas (Air) | Value |
|---|---|---|
| Diameter (μm) | — | 2–48 |
| Density (kg/m$^3$) | Ideal gas | 2933 |
| Cp (J/KgK) | Variable | 0.7 |
| Thermal conductivity (Kg/mK) | Kinetic theory | 0.314 |
| Molecular weight (kg/mol) | 0.028966 | 0.502 |

Figure 1b shows the sections of the modeled system and boundary conditions considered for the simulation of the air and particles passing through the nozzles. The inlet conditions were 300 °C and 5 bar, respectively. Such conditions are within the operational limits of the LPCS DYMET 423 system. Stationary and adiabatic walls were also considered as boundary conditions for the modeled nozzles. At the nozzle exit, atmospheric temperature and pressure were established.

On the other hand, a triangular arrangement was employed for meshing of the convergent, throat, and divergent zones of the nozzles. In addition, a rectangular array of 0.00018 m per element was selected for the prechamber and the outside atmosphere; a total range of elements and node values from 148,577 to 202,181 and 131,374 to 180,141, respectively, were used in the present study. The final element and nodes number was a function of the nozzle geometry studied. Additionally, the CFD results were compared with the analytic and 1D isentropic model from the compressible flow theory, which was used for the first time in a convergent-divergent nozzle by Dykhuizen & Smith (D&S) [19,23].

Experiments were performed with each of the nozzles in the LPCS equipment to measure the maximum temperature of the gas at two positions: at the exit of each nozzle

and at 10 mm away from the exit. Finally, a setup to measure temperature in real time at the two established positions was performed using a K-type thermocouple (Calor y Control, Querétaro, Mexico) connected to an Arduino® Uno board/Adafruit max31855 thermocouple amplifier array. The operating range of the thermocouple was −200 to 1350 °C, with a sensitivity of 0.25 °C. The maximum temperature values reported in this work corresponded to steady state experimental conditions [24].

In addition, BHAp particles were sprayed to obtain footprint-like deposits. A 316L stainless steel (SS) was employed as substrate material 2.54 cm × 3.00 cm × 0.6 cm in length, width, and thickness, respectively. The experiments were carried out at three different stand-off distances, namely 10, 20, and 30 mm. The powder feeding rate was 8 g/min, the substrate was heated at 200 °C, while the deposition time was 6 s. Furthermore, the temperature and gas inlet pressure were 300 °C and 5 bar, respectively, being the same as the spraying condition employed for the simulations. To evaluate the effect of the spraying parameters and performance of the conventional and modified nozzle geometries on the deposition behavior of BHAp powders, a Leica DMS1000 digital stereoscopic microscope (Leica, North Deerfield, IL, USA) was used to characterize the footprint-like deposits. Both footprint diameters and the deposited area were calculated from image analysis using the Image J® software (version 64-bit Java 8). An Olympus DSX510 equipment (Olympus, Tokyo, Japan) was used to obtain profilometry images of the footprints; this microscope performs a sweep in height, allowing a height profile of the surface to be obtained. BHAp coatings were obtained on 2.54 cm × 3.00 cm × 0.6 cm 316L SS coupons at a stand-off distance of 10 mm, on one spraying pass and with the same feeding rate and gas inlet parameters used for the footprints with both the conventional and modified nozzles. Cross-sectional images from the microstructure of the coatings were obtained from Scanning Electron Microscopy (SEM) using a JEOL 7610F microscope (JEOL, Tokyo, Japan) at 2 kV and 15 mm working distance.

## 3. Results and Discussion

### 3.1. Nozzle Behavior Modelling

Figure 2 shows the gas velocity and pressure profiles obtained for the conventional and modified nozzles as a function of the normalized axial distance (i.e., section position/length of section) obtained from CFD and 1D isentropic models, shown as a dotted line. Figure 2a presents the velocity profile for the nozzles obtained from both CFD and estimations from the 1D model. At the beginning of the convergent section (3) and at the end of the prechamber (2), an increase in velocity is observed for both nozzles. Additionally, this increase in velocity continues up to the beginning of the throat section (4). Particularly, the gas compression and acceleration process begins in the modified nozzle (red curve) at an axial distance of 0.0202 m from the beginning of the prechamber. In contrast, in the conventional nozzle the same process occurs at an axial distance of 0.02619 m from the prechamber. Furthermore, depending on the model, the behavior is different in the throat section. In the CFD case, an increase in velocity is still observed, while for the 1D model the velocity remains constant. This behavior would be related to idealizations considered in the 1D model, but not in the CFD model. As far as the divergent zone is concerned, the gas velocity continues to increase due to the gas expansion process. However, gas velocity fluctuations are observed in the CFD model, which would be related to the formation of shock waves inside the nozzle. These shock waves are formed due to the pressure difference between the inside of the nozzle and the external atmosphere. These fluctuations have been previously, both theoretically and experimentally, observed in LPCS systems [25,26]. Shock waves are often formed in convergent-divergent-type nozzles under expanded flow conditions, forming diamond-like patterns, which are associated with expansion and compression regions. Diamond patterns are dissipated out of the nozzles by viscous effects at a distance far from the nozzle exit, where the pressure takes the atmospheric value [27]. In the 1D model, the gas velocity increases, being higher than that obtained by CFD. The differences in both models are related to the idealizations of the 1D model, as

mentioned before, in addition to other phenomena, such as boundary layer formation and turbulence effects [23,28].

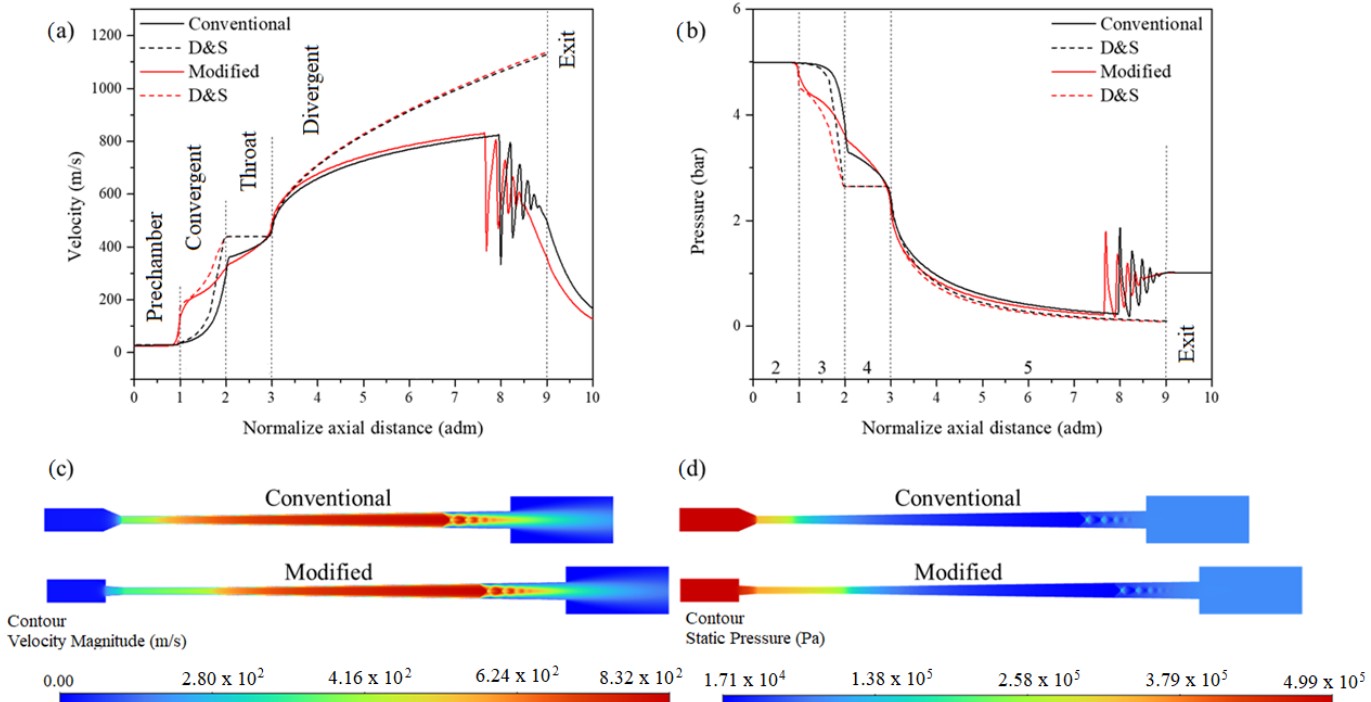

**Figure 2.** Velocity and pressure profiles for conventional and modified nozzles using CFD and 1D (D&S) models: (**a**) velocity, (**b**) pressure, and contour plots: (**c**) velocity and (**d**) pressure.

Figure 2b shows the variations in gas pressure for both nozzles, as well as their behavior in the 1D and CFD models. In both cases, the pressure tends to decrease along the nozzle. However, as mentioned for the gas velocity behavior, the gas pressure reduction seems to start at the prechamber section for the modified nozzle. At the same time, this process is observed in the converging section of the conventional nozzle. In addition, this change is more abrupt in the modified than in the conventional nozzle, which is related to its abrupt change in geometry. This behavior is observed in both models. In the throat section, the pressure remains constant for the 1D model, while for CFD results the pressure continues to decrease, even in the divergent section, as expected. In contrast, in the conventional nozzle the highest compression values are obtained at the end of the converging section and the beginning of the throat. This behavior promotes a smothered decompression process of the gas in the modified nozzle (red solid line in Figure 2a). Figure 2c presents the velocity contour for the two different nozzles in this work. One can see gradual changes in the gas velocity for the modified nozzle, which is consistent with the gradual gas decompression reported in Figure 2b,d. Yet, fluctuations are observed in the CFD model, which, as mentioned above, is related to shock waves.

The changes performed to the nozzle, from a conventional to modified design, have proved to have an influence on its performance. The latter promotes major changes in the internal gas flow, mainly at the transition zone between the prechamber and the nozzle inlet. Interestingly, the gas compression occurs earlier in the modified nozzle at the end of the prechamber and the beginning of the converging zone, in comparison with the conventional one. In the conventional nozzle, the highest compression value is obtained at the end of the converging section and the beginning of the throat. This fact promotes a smothered decompression process of the gas in the modified nozzle.

On the other hand, in the divergent section, Figure 2c,d suggests that gas velocity increases and gas pressure decreases after passing through the throat of each nozzle. This behavior is related to a supersonic type of expansion (i.e., Ma > 1). In addition, the results

reveal that the modified nozzle design can achieve slightly higher gas velocity values (1054 m/s) than those obtained for the conventional (1042 m/s) nozzle. This fact can be related to the longer diverging section and the pronounced geometric expansion ratio in the divergent zone of the modified nozzle. According to Buhl et al. [29], a long nozzle length can increase gas velocities, as confirmed in the present study when comparing both conventional and modified designs. However, the expansion ratio also contributes to gas acceleration. A short diverging length can be designed to keep high gas velocities by adjusting the expansion ratio, as observed in the comparison of the modified and conventional nozzles.

Figure 3 presents a velocity vectors plot for the nozzles in the prechamber and the convergent/front sections of the throat to appreciate the flow direction changes in these regions for both nozzles, resulting from the great difference in their geometries (i.e., cross-section area changes). In the conventional nozzle, the gas compression starts gradually from the beginning of the convergent zone to the throat, where the gas reaches a maximum compression value of 5.3 bar (see Figure 2b). In the modified nozzle, on the other hand, the gas compression occurs abruptly in a confined region at the beginning of the converging section. In addition, the presence of edges and 90° corners at the end of the prechamber in the modified nozzle promotes the formation of vortices and a velocity drop at the corners (arrows in Figure 3). The presence of forced vortices is common in fluids passing through certain surfaces with abrupt changes in sections. This causes the fluid to change its trajectory and rotate forcefully and generates a velocity drop that affects other variables, such as pressure [30]. As a result, the maximum compression pressure of the gas obtained in the modified nozzle was 5.7 bar (see Figure 2b). Interestingly, the gas compression occurs earlier in the modified nozzle, at the end of the prechamber and the beginning of the converging zone.

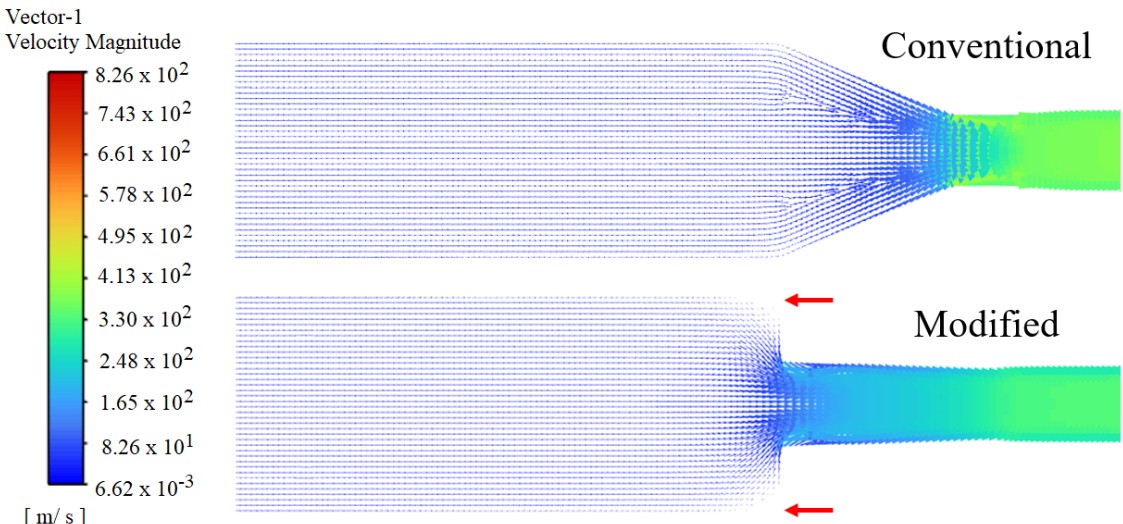

**Figure 3.** Velocity vector diagram for conventional and modified nozzles.

Table 2 shows the gas temperature values at the nozzle exit ($T_e$) and at a spray distance of 10 mm ($T_{10}$) following the methodology presented above. Table 2 also shows the gas temperature values ($T_e$) and ($T_{10}$) obtained from numerical simulations for the same operating conditions. The percentage difference between experimental and simulation data is also displayed in Table 2. However, that difference is smaller in the modified nozzle, with a maximum of about 4.77% with respect to the conventional nozzle. This fact can be associated with the more stable gas flow condition achieved at the measuring point of the experimental temperature value. That point is almost at the end of the diamond patterns observed in Figure 2 for the modified nozzle design. Interestingly, the measuring point of experimental temperature lies within the core of the diamond patterns observed in Figure 2

for the conventional nozzle, which makes the measurements difficult due to fluctuations. Huang & Fukunuma [31] also attribute differences between experimental and simulation velocity data to the overestimation resulting from mathematical solvers in CFD. Taking this information into account, the CFD model is considered adequate to describe the gas flow through the nozzles in a fairly approximate way, which is the reason for its success in the studies of metal particle deposition in HPCS and LPCS in the past [21,22,31].

**Table 2.** Gas temperature obtained from CFD simulations and experimental process.

| Condition | | Conventional | | | Modified | | |
|---|---|---|---|---|---|---|---|
| | | Meas | Simul | Δ (%) | Meas | Simul | Δ (%) |
| 300 °C 5 bar | $T_e$ (°C) | 268.2 | 281.38 | 4.68 | 282.0 | 282.91 | 0.32 |
| | $T_{10}$ (°C) | 259.2 | 290.43 | 10.75 | 273.5 | 287.20 | 4.77 |

$T_e$ = Exit, $T_{10}$ = 10 mm SoD.

### 3.2. BHAp Particles Deposition

The deposition parameters for producing HAp coatings were 300 °C and 5 bar for temperature and pressure gas inlet, respectively. The stand-off distances were 10, 20, and 30 mm. Under these conditions, results from CFD calculations indicate that BHAp particles arrived at the substrate with an impact velocity of 375, 359, and 340 m/s and 336, 318, and 299 m/s for the conventional and modified nozzles, respectively. In addition, the calculation of particle impact temperature led to a similar value of 270 °C for both nozzles. Therefore, the experiment not only demonstrated that both nozzles allow BHAp particles to be deposited, but they also allowed analysis of other parameters associated with nozzle design that influence particle deposition.

Figure 4 shows the results of the cold-sprayed BHAp prints obtained experimentally using both the conventional and modified nozzles. Footprint diameters increase with stand-off distance (Figure 4a), which could be related to the jet divergence at the outlet of the nozzle due to the gas expansion effect. Figure 4b exhibits the jet diameter calculated using CFD for the three stand-off distances used in this work (10, 20, and 30 mm). This result agrees with the experimental footprint diameter tendency. However, the simulated jet diameter has a different magnitude than the experimental footprint diameter obtained. Such a discrepancy may be associated with particle size, since it is possible that smaller particles, located at the edge of the jet, could be deflected by the bow shock. Additionally, the particles would present different trajectories depending on their diameter, which will be explained in more detail later in this section.

The estimations of velocity magnitude from simulation results (Figure 4c,d) and optical micrographs from experimental BHAp prints (Figure 4e,f) were analyzed for both nozzles. Interestingly, the gas velocity magnitude was higher in the conventional than the modified nozzle for the three spraying distances tested. The highest velocity achieved by the gas stream is observed at the jet centerline and decreases exponentially as a function of its diameter. This behavior is expected because the jet interacts with the surrounding atmosphere, giving rise to entrainment and turbulence phenomena that substantially reduce the jet velocity. Figure 4c,d show gray areas under the magnitude velocity vs. radial distance curves, representing the experimental print diameter obtained for both nozzles. This result shows a limit of gas magnitude velocity (85–100 m/s) from which deposition of BHAp particles does not occur. Figure 4e,f present the optical micrographs of the BHAp footprints obtained in this work. The conventional nozzle produces a donut-like morphology (Figure 4e), while the footprints obtained using the modified nozzle present a whole area deposited with BHAp (see Figure 4f). Figure 5 shows profilometry images from the footprints obtained. Figure 5a displays the profiles of heights across the footprints prepared with the conventional nozzle. One can see that accumulation of BHAp particles is observed at the outer diameter of the footprints (see red arrows), while a decrease in the deposited particles is observed at the center of the footprints (see blue arrows). Figure 5b

displays the height profile images from the footprints prepared with the modified nozzle. One can see that peaks are distributed across the footprints and there is no accumulation of particles in a specific region. These experimental observations suggest that the conventional nozzle tends to concentrate a higher volume of particles around the centerline of the jet, which is promoted by its high velocity. The high volume of particles impacting at high velocity promotes a decrease in the deposition of particles at the center of the circular area on the substrate, where the jet containing the BHAp particles arrives, forming a donut-like print. On the other hand, in the modified nozzle, such particle volume concentration does not occur because of the lower jet velocity achieved in comparison with the conventional nozzle. This allows particles to disperse across a higher fraction of the radial diameter of the jet. As a result, accumulation of particles at the outer diameter does not become evident in the footprints obtained with the modified nozzle. This suggests that particle dispersion occurs around the centerline of the jet and reveals that if the dispersion is not enough, such volumetric concentration of particles can affect the deposition efficiency of the particles. Although the BHAp particles were dispersed differently in the two nozzles, the external diameters of the prints were similar, suggesting a minimum impact velocity value for the BHAp particle deposition. The velocity value was calculated from simulation results ranging between 80 m/s and 253 m/s. Further experimental studies are encouraged to validate the "minimum impact velocity" concept through statistical analysis under different spraying conditions.

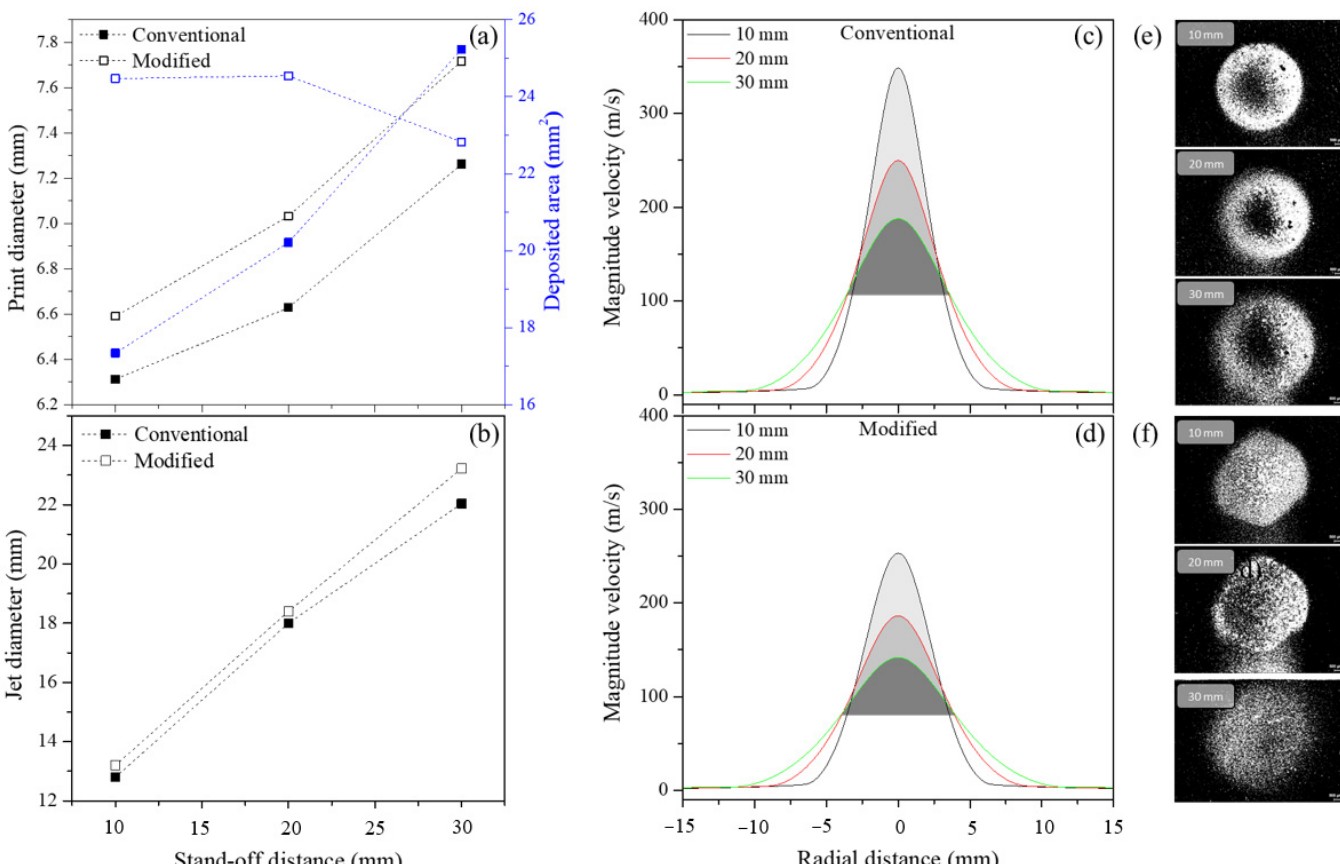

**Figure 4.** Summary of the footprints and jet diameters: (**a**) experimental footprint diameter and deposited area, (**b**) jet diameter obtained with CFD, jet velocity profile using the (**c**) conventional and (**d**) modified nozzles. Optical micrographs of the footprints are shown in (**e**) for the conventional nozzle and (**f**) for the modified nozzle.

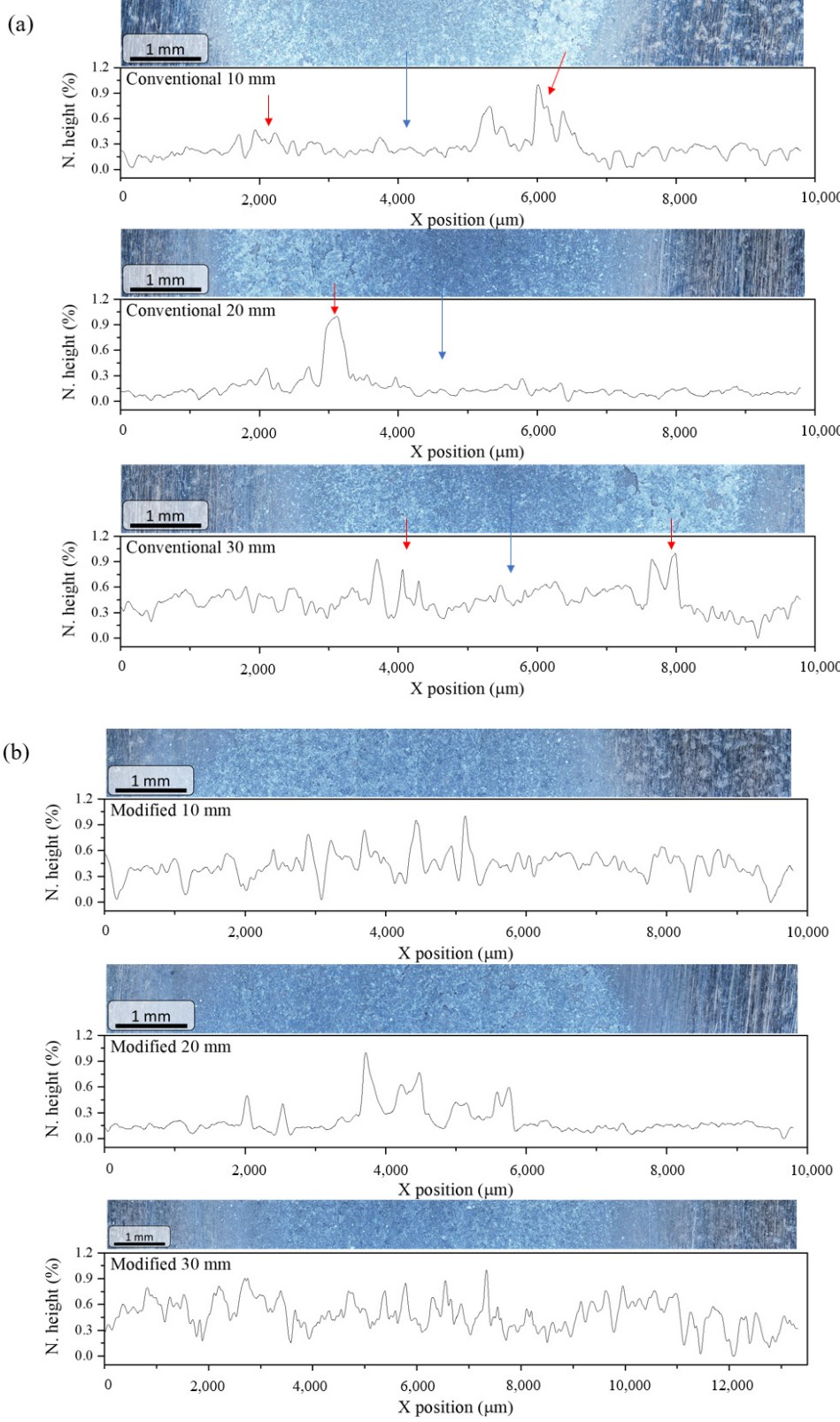

**Figure 5.** Height profile images from the footprints obtained: (**a**) using the conventional nozzle, (**b**) using the modified nozzle. The red arrows display accumulation of deposited HAp while the blue arrows display regions with a smaller number of deposited particles. N. height is the normalized height value with respect of the maximum height (see Supplementary Material, Table S1).

Figure 6 shows the cross-sectional view of BHAp coatings prepared with the conventional and the modified nozzles at a stand-off distance of 10 mm. The BHAp coatings had a thickness of 10 μm and 17 μm, respectively. These coatings presented microstructural characteristics observed in previous studies for cold-sprayed agglomerated BHAp coatings [13] such as fractures and compacted particles that result from a pore collapse mechanism. This mechanism consists in particles continuously impacting on the substrate, generating a compaction process in previously deposited particles. This process is followed by a dynamic fragmentation process, which is produced by cracking and crushing [10,13]. Interestingly, BHAp coatings obtained by employing the modified nozzle had a more homogeneous thickness when compared against those prepared employing the conventional counterpart.

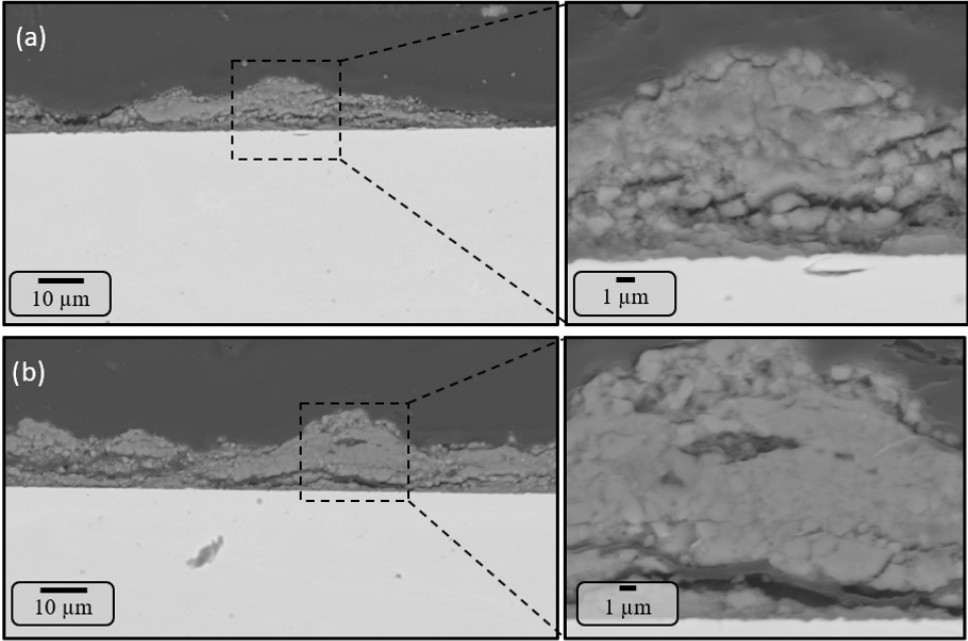

**Figure 6.** SEM images from the cross-section of BHAp coatings obtained at 10 mm—300 °C—5bar —one spraying pass (**a**) with the conventional nozzle, (**b**) with the modified nozzle.

Figure 7 shows the particle temperature as a function of its velocity and kinetic energy at impact when sprayed using the conventional and modified nozzles. The particle temperature ranges from 250 to 276 °C. Figure 7a,b display two temperature ranges that show dependency on the particle size. The first range lies between 2 to 10 μm, where the temperature tends to increase, because the smallest particles are highly affected by the flow and have less residence time in the gas. Therefore, their temperatures would be lower. The second range corresponds to particles with a diameter greater than 15 μm, in which the particle temperature tends to decrease as the diameter of the particles increases, since larger particles are more difficult to be heat-affected, as expected. On the other hand, the particle temperature as a function of its impact velocity is shown in Figure 7a, where it is observed that larger particles arrive at the substrate with a lower velocity, as they are more difficult to accelerate. Nevertheless, those particles have greater inertia and tend to keep their velocity while flying in the gas stream. Figure 7b shows the particle temperature as a function of the kinetic energy when reaching the substrate. As the particle size increases, so does its kinetic energy. This result is related to the following equation $E_k = 1/2(\pi dp^3)\rho v^2$; in this expression, the diameter of the particle has a cubic dependence, whereas that of the velocity is quadratic. That is why the diameter of the particle has a more significant influence on the increase in kinetic energy compared with the velocity.

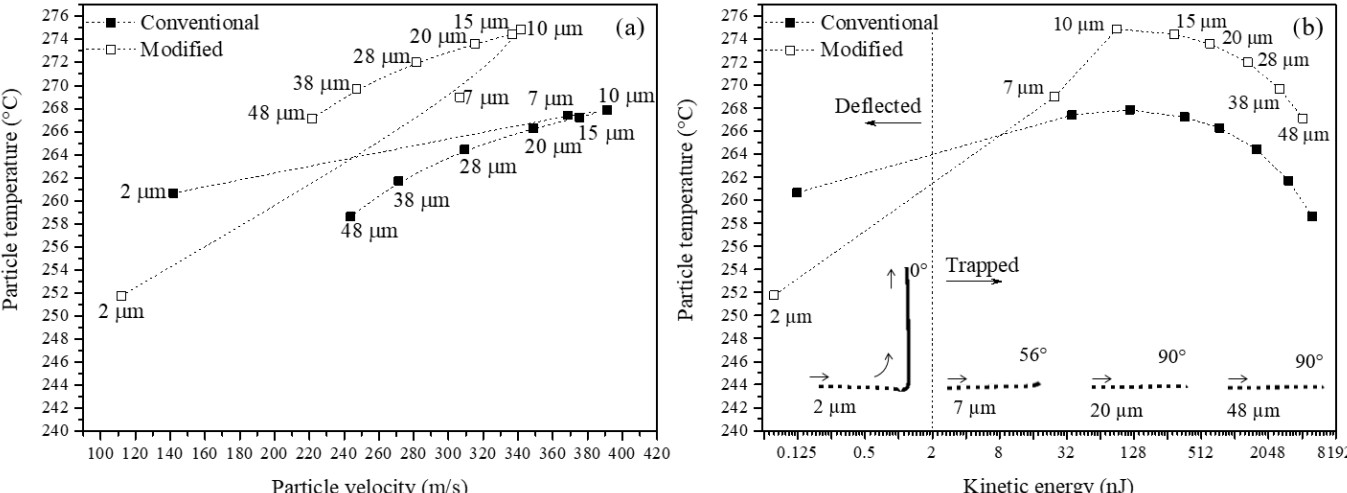

**Figure 7.** Conditions of the particle upon arrival at the substrate as a function of its diameter (**a**) particle temperature as a function of particle velocity and (**b**) particle temperature as a function of the kinetic energy of the particle.

Furthermore, the same zones identified before are also used to explain the particles' behavior as a function of the kinetic energy (Figure 7b). In the first zone, corresponding to particles smaller than 10 μm, the kinetic energy is in an order of magnitude of a few nJ. In the second zone, corresponding to particles larger than 15 μm, the kinetic energy increases up to four orders of magnitude. The particle trajectory behavior when interacting with the substrate is also a function of particle size (Figure 7b). Particles smaller than 2 μm are deflected by the bow shock effect, following the gas trajectory, i.e., moving parallel to the substrate (0°) due to their low inertia. Therefore, particles between 2 and 10 μm manage to reach the substrate. However, those particles arrive at an angle different than 90°, possibly rebounding rather than being deposited. Angular impact is known in thermal spray for having a poor contribution to coatings formation, because tangential velocity detaches particles from the surface [32,33]. Finally, particles larger than 15 μm reach the substrate surface at an angle of 90°, which is associated with greater inertia and kinetic energy, so these may be the particles that participate in the coating formation.

Some works have proposed mechanisms for HAp deposition, such as those presented by Cinca et al. [13] and Chen et al. [10]. The approach presented in this work allowed the completion of the previously proposed observations by those authors. They prepared HAp coatings by cold spray using particle sizes with an average diameter of 47 μm, coating thicknesses of up to 100 μm, and minimal porosity. Those results should be attributed to the use of large particles, which reduced the effect of the angular impacts caused by small particles that have a poor contribution on the crushing and densification process of the deposited layers.

Moreover, according to Ravanbakhsh et al. [34], there may be an energy balance between the impact kinetic energy ($E_k$) and fracture energy of the particles ($E_s$). The latter would be expressed as $E_s = 1/2\pi d^2\gamma$; where "γ" is the surface energy and "d" is the diameter of the particle. The authors also suggested that this relationship must be greater than one to build up a coating, i.e., when the impact energy of the particle is greater than the energy required for fracture. Therefore, larger particles could better meet this criterion than smaller ones. Elsenberg et al. [35], however, suggest that there is a sufficiently large particle size that, instead of promoting deposition, generates surface erosion. In addition, the nature of the deposited particle could promote phenomena ranging from particle fracture to viscous behavior. In summary, porous agglomerated BHAp particles such as those studied in this work and by other authors [10], would have an average particle size of 40 μm to promote the formation of a coating. In fact, these conditions were also favorable

in the work of Chen et al. [10], allowing the particles to gain enough kinetic energy to reach the substrate and promote a thick and dense coating.

## 4. Conclusions

In recent years, cold-sprayed bioceramic coatings have been of great interest in the scientific community due to the low processing temperatures achieved in the process, which can give some key advantages to bioceramic coatings over those produced by conventional thermal spray. Therefore, the present study aimed to better understand the deposition of BHAp particles by cold spray. Simulation tools, nozzle geometry, and experimental work were carried out to accomplish this goal using BHAp powders. This approach resulted in the following conclusions:

- Nozzle design plays a key role in depositing BHAp powders in the LPCS process. Experimental and CFD analysis revealed that nozzle geometry leading to a high concentration of particles around the centerline of the jet produce non-homogenous coatings under the conditions experienced in this work. This fact can be promoted due to the formation of a donut-like footprint of powder, which reduces the deposition rate of powder on the substrate surface. Hence, a good-quality HAp coating may be obtained by employing a nozzle having a radial jet velocity profile with a smaller exponential decrease than that found for the conventional nozzle used in this work.
- Particle-size distribution also plays an important role in depositing BHAp powders. Simulations suggest that particles smaller than 10 µm can be deflected out of the centerline of the jet and may not contribute to the formation of the coatings. Particles larger than 10 µm can experience different degrees of deceleration. The larger the particle diameter, the lower the impact velocity and temperature. These particles can contribute to the formation of coatings since they keep their trajectories while maintaining the probability of cracking and crushing. In fact, the morphology of deposited coatings obtained in this work revealed microstructural characteristics such as cracking and compaction of agglomerated particles.
- The present study presents simulations and experimental results using a conventional and a modified nozzle geometry. Further research from a fluid dynamics point of view and experimental validation could be performed in the future to explore different radial jet velocity profiles. Considering the findings in this work, it is likely that optimized nozzle geometries can be developed in the future for the preparation of BHAp coatings.

**Supplementary Materials:** The following supporting information can be downloaded at: https://www.mdpi.com/article/10.3390/coatings12121845/s1, Figure S1: SEM images about morphology of BHAp particles. (A) General view of the BHAp powder; (B) cross-sectional view of BHAp particles showing that are spherical and agglomerated; Figure S2: Experimental particle size distribution of BHAp powders; Table S1: Heights of the footprints measured by profilometry.

**Author Contributions:** Formal analysis, P.A.F.-S., A.L.G.-B., J.H.; Funding adquisition, J.H., C.A.P.-S., J.C.-C., A.L.G.-B.; Investigation, P.A.F.-S., A.I.G.-P., E.M.R.-V.; Writing—original draft, P.A.F.-S., J.H., A.L.G.-B., Writing—review & editing, J.H., P.A.F.-S., A.L.G.-B., C.A.P.-S. All authors have read and agreed to the published version of the manuscript.

**Funding:** This research was funded by National Council of Science and Technology of Mexico (CONACYT), the program "Investigadores por Mexico" projects 848 and 881, and to the Ciencia de Fronteras program "Paradigmas y controversias 2022" project number 320126.

**Institutional Review Board Statement:** Not applicable.

**Informed Consent Statement:** Not applicable.

**Data Availability Statement:** Not applicable.

**Acknowledgments:** The authors acknowledge the National Council of Science and Technology of Mexico (CONACYT) also acknowledge the National Laboratory of Thermal Spray of Mexico (CENAPROT), LIDTRA national laboratories from Cinvestav-Queretaro and the Additive Manufacturing Network (CONMAD-CIDESI) for allowing the use of their thermal spray and characterization facilities.

**Conflicts of Interest:** The authors declare that there is no conflict of interest regarding the publication of this paper.

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
