# Peer review of "Nozzle Geometry and Particle Size Influence on the Behavior of Low Pressure Cold Sprayed Hydroxyapatite Particles"

_coatings, doi:10.3390/coatings12121845_

Round 1
Reviewer 1 Report
Please see the attachment.

Author Response
Dear Reviewer,
Thank you for your comments. Please, see the attachment with the response to the comments.

Reviewer 2 Report
Interesting study, a few comments for further improvements:
Line 25-27: Not clear what authors trying to convey. Please rewrite.
Line 36: CS temperature can go upto 1100 C using high-pressure and high-temp system such as Impact 5/11
Line 40: pressure can be upto 60 bar using state-of-the-art CS systems
Line 45-56: Unnecessary information
Line 112: make a separate paragraph from "The present study seeks..............
Table 1: What is Aire?
Figure 2c,d, Figure3: Increse font sizes within the figure, some are very small and not readable at present
Figure 4: Check caption, line 332
General comment: the article can be written in a more concise manner, where appropriate.
Author Response
Dear Reviewer,
Thank you very much for your valuable comments. Please, see the attachment with the response to the comments.

Round 2
Reviewer 1 Report
Now this paper is significantly better and can be recommended for publication.